# Insulin Receptor Trafficking: Consequences for Insulin Sensitivity and Diabetes

**DOI:** 10.3390/ijms20205007

**Published:** 2019-10-10

**Authors:** Yang Chen, Lili Huang, Xinzhou Qi, Chen Chen

**Affiliations:** School of Biomedical Sciences, The University of Queensland, Brisbane 4072, Australia; yang.chen1@uq.edu.au (Y.C.); l.huang2@uq.edu.au (L.H.); xinzhou.qi@uq.net.au (X.Q.)

**Keywords:** insulin receptor, endocytosis, trafficking, recycling, insulin resistance

## Abstract

Insulin receptor (INSR) has been extensively studied in the area of cell proliferation and energy metabolism. Impaired INSR activities lead to insulin resistance, the key factor in the pathology of metabolic disorders including type 2 diabetes mellitus (T2DM). The mainstream opinion is that insulin resistance begins at a post-receptor level. The role of INSR activities and trafficking in insulin resistance pathogenesis has been largely ignored. Ligand-activated INSR is internalized and trafficked to early endosome (EE), where INSR is dephosphorylated and sorted. INSR can be subsequently conducted to lysosome for degradation or recycled back to the plasma membrane. The metabolic fate of INSR in cellular events implies the profound influence of INSR on insulin signaling pathways. Disruption of INSR-coupled activities has been identified in a wide range of insulin resistance-related diseases such as T2DM. Accumulating evidence suggests that alterations in INSR trafficking may lead to severe insulin resistance. However, there is very little understanding of how altered INSR activities undermine complex signaling pathways to the development of insulin resistance and T2DM. Here, we focus this review on summarizing previous findings on the molecular pathways of INSR trafficking in normal and diseased states. Through this review, we provide insights into the mechanistic role of INSR intracellular processes and activities in the development of insulin resistance and diabetes.

## 1. Overview of Insulin Receptor (INSR) Signaling Regulation

INSR plays essential roles in fundamental biological processes including proliferation and energy metabolism [1]. INSR activities are often dysregulated in metabolic disorders such as diabetes. The abnormality of INSR function in the pathogenesis of diabetes has highlighted the importance of understanding the comprehensive regulation of INSR and its downstream signaling events. INSR is a member of the receptor tyrosine kinase (RTK) family [2] and its basic trafficking and regulation have been established; however, little is known about the underlying molecular mechanisms of these events.

Ligand-stimulated endocytosis of RTKs has been consensually considered as a process of signal termination to modulate the intensity and duration of receptor activities. This concept is supported by the prolonged insulin signaling in cells without INSR endocytosis [3,4] and the targeted lysosomal degradation of internalized RTKs. Insulin resistance, a major component in metabolic disorders, is generally viewed as a post-receptor signaling defect where INSR activities remain unimpaired. It has been extensively reviewed for INSR downstream signaling pathways such as PI3-kinase (PI3K)/AKT and Ras/Raf/MEK/ERK [5] and their alterations under insulin resistance state [6]. However, emerging studies are demonstrating a far more complicated picture of INSR. INSR shows spatial preference in activating its downstream signaling by initiating the PI3K/AKT pathway on the plasma membrane and activating Ras/Raf/MEK/ERK pathway when internalized [3,7].

This review summarizes the regulation of INSR trafficking, the underlying mechanisms under normal conditions, and the altered INSR trafficking under insulin resistance and diabetic conditions.

## 2. Ligand-Dependent INSR Activation

As an important integral membrane protein, INSR is expressed in all mammalian cells and is regulated at multiple stages through the innate intracellular activities. The normal metabolic fate of INSR follows endocytosis (internalization and endocytic trafficking), sorting, endosomal trafficking, and recycling or degradation.

INSR endocytosis is predominantly driven by ligands and accompanied by a low rate of constitutive internalization [8]. The endogenous ligands of INSR include insulin, insulin-like growth factor 1 (IGF-1) and IGF-2 [9]. Ligand-induced autophosphorylation is essential for INSR internalization in all cells except hepatocytes, where significant constitutive internalization is identified [10]. 

Once activated, INSR autophosphorylates and is internalized into the cell through endocytosis. Phosphorylated INSR recruits and activates target molecules including insulin receptor substrates (IRSs), Src homology 2-B (SH2-B), and protein phosphatases to trigger downstream signaling events [6]. INSR is then regulated in early endosome (EE), a protein-sorting platform, for subsequent trafficking [11]. After being inactivated and sorted in EE, the majority of INSRs are recycled back to the plasma membrane with a small proportion translocated to late endosome for degradation or into the nucleus [12,13,14] (Figure 1). 

## 3. Ligand-Dependent INSR Endocytosis: A Spatial Modulator of INSR Signaling

Endocytosis is an important mediator of INSR activities. Endocytosis of INSR contains two major steps, namely, internalization of INSR from the plasma membrane (internalization) and relocation of INSR into early endosomes (endocytic trafficking). Endocytosis effectively mediates INSR availability on the plasma membrane and provides spatial regulation of INSR downstream signaling pathways [15]. Activation of PI3K/AKT pathways by INSR begins at the plasma membrane and terminates via endocytosis. On the other hand, intracellular initiated signaling pathways (e.g., the Ras/Raf/MEK/ERK pathway) can be activated during endocytic trafficking of INSR [7]. 

Regulation of INSR endocytosis involves two protein-coupled endocytic pathways—clathrin-dependent endocytosis and caveolar endocytosis (also known as clathrin-independent). The clathrin-dependent endocytosis was initially found in hepatocytes [16]. Until recently, caveolar endocytosis of INSR was identified in several types of cells, such as adipocytes and pancreatic beta cells [3,17]. Therefore, the two INSR endocytosis pathways are likely to be tissue-dependent and alternation in either pathway may be involved in the pathogenesis of type 2 diabetes mellitus (T2DM). There is currently limited evidence suggesting whether these distinct pathways could lead to distinct intracellular processing of INSR, either to recycling or to degradation. The endocytosis of INSR in this review will focus primarily on hepatocytes, adipocytes, and pancreatic beta cells.

### 3.1. Clathrin-Mediated INSR Endocytosis

Clathrin-dependent endocytosis is mainly mediated by clathrin, adaptor protein 2 (AP2) complex, and clathrin-associated sorting proteins (CLASPs) [18]. AP2 serves as a hook and fixes the cargo to the plasma membrane for internalization [19]. CLASP is involved in cargo recognition to help classify cargos for subsequent sorting events [20]. In hepatocytes, INSR internalization requires autophosphorylation of regulatory tyrosine 1146, 1150, and 1151 to redistribute INSR on the plasma membrane from microvilli to clathrin-coated pits for internalization [21]. Anchoring INSR on clathrin-coated pits is supported by a juxta-membrane cytoplasmic segment on the INSR beta subunit with an NPXY sequence, a typical sequence required for clathrin-mediated internalization [21,22,23]. AP2 was then recruited to anchored INSR by spindle checkpoint proteins mitotic arrest deficient 2 (MAD2) and BUB1-related protein 1 (BUBR1) [24]. Depletion of MAD2 and BUBR1 can induce INSR membrane retention and increase insulin sensitivity by enhancing AKT signaling [24]. 

CLASPs include a number of proteins and yet little is known about individual CLASPs for INSR endocytosis. A recent study suggests that limb region 1 (LMBR1) protein homolog can be one of the potential CLASPs regulating INSR endocytosis [25]. Liver-specific knockout of LMBR1 domain containing 1 (LMBD1) in murine shows that the clathrin-dependent endocytosis was attenuated with enhanced receptor signaling, which may be related to the delayed internalization of INSR from the plasma membrane [25]. β-site amyloidogenic cleavage of precursor protein-cleaving enzyme 2 (BACE2) is another candidate CLASP for INSR endocytosis [26]. Immunofluorescence analysis demonstrates that BACE2 colocalizes with insulin in clathrin-coated vesicles on the plasma membrane in MIN6 cells, a pancreatic beta cell line. When inhibited, BACE2 accumulates in clathrin-coated vesicles accompanied by decreased insulin internalization [26]. Alterations in INSR recruitment, deficient AP2, and CLASP may all potentially lead to ineffective identification of insulin-INSR complex and potentially delay the process of both endocytosis and protein sorting. Such alterations may exaggerate insulin response in the short term due to INSR retention on the plasma membrane, but may lead to a long-term decrease in INSR membrane expression due to reduced INSR recruitment and development of insulin resistance.

### 3.2. Clathrin-Independent INSR Endocytosis 

Caveolar endocytosis is a type of clathrin-independent endocytosis and generally slower than clathrin-mediated endocytosis. Caveolar endocytosis of INSR has been shown in adipocytes and pancreatic beta cells [27]. Microscopic colocalization analysis discovered that INSR colocalized with caveolin in adipocyte during internalization [17]. A further study specified that in pancreatic beta cells, the INSR internalization was mediated by caveolin-1 [3].

The internalization of INSR via caveolar endocytosis is regulated by caveolins and the lipid rafts on the plasma membrane [28]. INSR autophosphorylation triggers caveolin-1 phosphorylation at tyrosine-14 to initiate endocytosis [17,29]. It has been reported that caveolin-2 also regulates the activity of INSR [30]. Alternative translation initiation of caveolin-2 desensitizes INSR and target INSR for degradation [30]. In adipocytes, INSR locates within caveolin-enriched membrane microdomains, or lipid rafts [31]. Lipid imbalance in lipid rafts may impair INSR activities and subsequent intracellular signaling [31,32]. Cholesterol and sphingolipids are main components of lipid rafts. Inhibition of cholesterol biosynthesis perturbs lipid raft and attenuates INSR activities [33]. GM3, synthesized from sphingolipid, was shown to compete with caveolin in binding to INSR and relocated INSR from caveolin-enriched microdomains to areas lacking caveolin [34]. Such relocation stopped INSR from being internalized and enhancing INSR signaling on the plasma membrane [35].

### 3.3. Candidate Regulators of INSR Endocytosis

Factors directly involved in endocytosis are not the only regulators of INSR internalization. Several other candidate regulators have been proposed. Embryonic fibroblast from PKC epsilon knockout mice demonstrated attenuated INSR internalization upon insulin stimulation. Such attenuation is associated with reduced expression in CEACAM 1, a receptor substrate modulating insulin clearance [36]. Loss of function mutation of Rabs indicated that Rab5 was required for INSR internalization and signaling in hepatocytes [37]. Moreover, extracellular matrix participates in the internalization of INSR. A study has found that a proper actin organization is critical for INSR internalization and suggested that function of insulin receptor can be altered when cells adhere to different extracellular matrices [38]. It is reasonable to believe that disturbance of INSR endocytosis may affect INSR sensitivity or cause insulin resistance. By studying the endocytosis of INSR, potential therapeutic targets may be identified to overcome insulin resistance.

## 4. INSR Endosomal Sorting: The Fate of INSR

Receptor sorting determines the subsequent processing of internalized INSR, namely, recycled back to the plasma membrane, sent to lysosome for degradation, or relocated to organelles like trans-Golgi apparatus and nucleus. The existing mechanism for RTK post-endocytic sorting is deciphered by studying epidermal growth factor receptors [39]. A current model suggests that the sorting of ubiquitinated RTKs is mediated by endosomal sorting complexes required for transport (ESCRT) [40,41,42]. Receptor-associated proteins and ligands are disassociated from RTKs for further receptor processing [43]. EE serves as the sorting platform during this process. 

INSR remains activated in EE until dephosphorylated by ligand dissociation [44]. A complex regulatory process decides the controvert destiny of INSR by selectively dephosphorylating tyrosine on INSR via protein tyrosine phosphatase (PTP) [45]. INSR dephosphorylation is a rapid reversible process in insulin sensitive tissues [46,47,48] and a crucial step for INSR recycling to the plasma membrane [49]. In adipose tissue and liver, INSR activity increases transiently following its dephosphorylation [50,51]. Enhanced INSR activity includes facilitated endocytosis and the recycling process to promote signal activation and rapid removal of excessive insulin from circulation. Several proteins are found to be potentially involved in INSR sorting. A patient study and animal models have suggested that Bardet–Biedl syndrome proteins are necessary to localize INSR onto the cell surface [52]. Another possible protein involved in INSR sorting is APS, an adaptor protein mediating the ligand-stimulated ubiquitination of INSR [53]. APS-mediated INSR ubiquitination may facilitate INSR internalization without increasing degradation, suggesting its potential role in sorting INSR into the recycling process [54]. INSR isotype A can be internalized through clathrin-dependent and -independent pathways upon ligand stimulation, but only the clathrin-dependent internalization is required for INSR-A degradation [55], suggesting distinct sorting processes are involved. 

Interestingly, sorting may be enhanced or simplified for INSR in liver under specific conditions. Postprandial insulin secretion is biphasic with stimulated acute first-phase insulin secretion and followed by a prolonged second-phase insulin secretion. A large amount of insulin is secreted and cleaned from the circulation in the first half hour after a meal during the first-phase insulin secretion [56]. The intense requirement for rapid insulin removal by hepatocytes is assisted by insulin receptor recycling within the first hour, which involves a rapid sorting process to achieve a high efficiency [57]. This process is likely to involve a bulk recycling pathway upon insulin stimulation, which initiates from early endosomes without going through the endosomal sorting compartment [58]. 

### 4.1. INSR Trafficking

Receptor trafficking contains two stages—endocytic trafficking and endosomal trafficking. Endocytic trafficking transfers the internalized receptors to EEs. After INSR is sorted in EE, endosomal trafficking takes place to control the relocation of INSR to intraluminal vesicles such as lysosome and the endosomal recycling compartment. 

Rabs are important regulators of endosomal INSR trafficking and fusion. Rab5 directs the transport and fusion of endocytic vesicles to and with EEs, whereas Rab4 is thought to control protein trafficking from EEs back to the plasma membrane [59]. A null function mutation experiment of Rab5 suggests that Rab5 is essential for INSR endocytosis and signaling [37]. Although there is lack of knowledge on the interaction between INSR and other proteins in the Rab family, several models have recently been proposed to elaborate the underlying mechanism between INSR and Rabs [60]. Further studies are required to testify proposed models. Due to the distinct responses and requirements for insulin, different cell types may prefer specific types of INSR trafficking. In hepatocytes and adipocytes, INSR is trafficked through the recycling path upon insulin stimulation as to maintain sufficient cellular responses to insulin. In pancreatic beta cells, INSR trafficking bypasses Rab5a, Rab7, or Rab11a [3], which are involved in RTK recycling processes. It suggests that INSR may be targeted for degradation after insulin stimulation in beta cells. 

Sorting nexin (SNX) is a family of cytosolic proteins with membrane association and protein sorting potentials [61]. Immuno-coprecipitation showed that SNX1, SNX2, and SNX4 interacted with INSR [62]. Functional analysis suggests that these proteins are likely to be parts of the INSR trafficking process [62]. SNX9 is found to regulate the processing and trafficking of INSR and steers insulin action [63]. 

Other than regulation by proteins in Rab and SNX family, INSR trafficking is modulated by endosomal pH, intermediate filaments, and ion transporters. Endosomal pH plays an important role in endosome trafficking. An acidic intra-endosomal pH is required for endocytosis and trafficking of INSR [64,65]. In hepatocytes, epithelial cell intermediate filaments, keratins 8 and 18, can regulate INSR signaling via the INSRS1/PI3K/Akt pathway and microtubule trafficking [66]. As an ion transport, zinc transporter 14 (Zip 14) influences INSR trafficking in liver. In Zip14 knockout mice, activities of the zinc-dependent insulin-degrading proteases are impaired due to low cytosolic zinc levels [67]. INSR remains phosphorylated and trapped in EEs, which indicates that endosomal trafficking of INSR is inhibited [67]. 

### 4.2. Regulation of INSR Recycling

RTKs are generally targeted for degradation after activation [39]. However, the majority of INSR is preserved and reused through the recycling process, especially in liver and adipose tissue. Liver is the major organ for insulin clearance [68], which demands a large amount of INSR. In hepatocytes, studies have shown that 60% of labelled INSR is recycled back to the membrane in five hours after insulin treatment, signifying recycling as a major and rapid process of INSR maintenance [57,69,70,71,72,73]. In adipocytes, INSR recycling seems to be much quicker than hepatocytes with recycled INSR found back onto cell surface in six minutes and little or no loss of labelled INSR is found in four hours after insulin stimulation [74,75]. The biological significance of INSR recycling is not only to facilitate insulin removal but to enhance short-term insulin signal as well. Recycled INSR shows stronger activities than unrecycled ones, which may contribute to the rapid clearance of insulin from circulation after a meal [76,77]. Increased INSR internalization process does not necessarily lead to elevated receptor degradation [14]. It is only when INSR is prevented from recycling by chloroquine that it is trapped intracellularly and targeted for a degradation pathway [75]. While the detailed mechanism of INSR recycling is unknown, it is reasonable to refer to the established recycling model of RTK. In this model, rapidly recycling of ligand-receptor complexes are from peripheral EE with a slower recycling from pericentriolar endosomes [78], which are regulated by Rab4 and Rab11, respectively [79]. Specific “recycling” sequence motifs required by other classes of receptors have not been found in RTKs [80]. Although a clear INSR recycling model is still missing, abundant evidence promotes fundamental hypothesis. Factors modifying INSR endocytic trafficking may be involved in INSR recycling. Both endocytosis and the recycling process of INSR are insulin dose-, temperature-, and pH-dependent, which favors an acidic environment [72,81]. Inhibition of INSR endocytic trafficking leads to dose-dependent elevated INSR residency and phosphorylation within the endosomal recycling compartment [82]. 

### 4.3. Nuclear Translocation of Internalized INSR

Lysosomal degradation and recycling to the plasma membrane are fundamental regulators to decrease and increase INSR signaling, respectively. Besides these two regulators, internalized INSR can be relocated to the nucleus. Despite being known for decades, INSR trafficking to the nucleus is often overlooked and the underlying mechanism remains controversial. Not until recently was a novel nuclear location sequence identified for INSR nuclear transfer [83]. In addition, it has been reported that post-translation modifications like receptor ubiquitin-like modifier (SUMO)-ylation and phosphorylation are required for effective nuclear translocation [84,85]. 

## 5. INSR Trafficking and Diabetes

### 5.1. Alteration of INSR Trafficking in Insulin Resistance

Alteration in post-receptor metabolic pathways is not the only cause of insulin resistance. The response to insulin can be influenced by the availability of INSR on the plasma membrane, INSR trafficking processes, and the balance between these processes. Downregulation of INSR has been identified in adipose tissue of both obese animals and patients, which is associated with insulin resistance [86]. Receptor downregulation is one of the mediators to control the availability of INSR. Intracellular trafficking of INSR, including endocytosis and recycling, are all monitoring the amount of INSR on the plasma membrane. Alteration of these activities will lead to an imbalance of INSR availability, which can further develop into insulin resistance (Figure 2).

### 5.2. Alterations of INSR in Insulin Resistance

Emerging evidences suggest that disruption of INSR can directly result in insulin resistance [11]. Genetic mutations in INSR can lead to conditions with severe insulin resistance (Table 1). Besides mutation on INSR, patients with T2DM have shown an impaired rate of insulin internalization compared with that in T1DM patients with normal insulin sensitivity. With the distinct characteristic of insulin resistance in T2DM pathology and the impaired INSR internalization occurring in T2DM patients, it is not surprising to anticipate a cause–effect relationship between INSR trafficking and insulin resistance [87]. Failure of INSR internalization leads to prolonged insulin existence in circulation causing hyperinsulinemia and further aggregates insulin resistance as shown in patients with an arginine to cysteine 252 mutation in INSR [88]. 

As shown in both in vivo and in vitro studies, disruption of INSR endocytosis may contribute to insulin resistance. Impairment of clathrin-dependent INSR endocytosis has been identified in the hepatocytes from hyperinsulinemia mice [125]. The role of impaired functionality of lipid rafts and caveolins has been reviewed in the development of insulin resistance, respectively, through alteration of INSR endocytosis [28,126,127]. Impaired INSR endocytosis due to increased viscosity and disrupted lipid orientation of the plasma membrane exerts a direct effect on decreased insulin action and development of insulin resistance. The alteration of caveolar endocytosis can contribute to insulin resistance in patients with abnormal amounts of GM3 in serum during the development of metabolic syndromes [128]. Abnormal production of GM3 was found in obese patients closely related to their high risk of insulin resistance and diabetes [129]. In addition, the role of cholesterol has been extensively studied in the pathogenesis of insulin resistance [130]. Accumulating evidence has indicated the importance of caveolin family proteins in the pathology of insulin resistance. Caveolin-1 null mice under a high-fat diet developed similar metabolic disarrangements, including insulin intolerance and postprandial hyperinsulinemia, to those in prediabetic patients [131]. Recent studies have confirmed that short-term exposure to high glucose significantly decreased the binding affinity of insulin to caveolin-1 and INSR in mature adipocytes, which might contribute to the development of insulin resistance in early T2DM [132]. Reduced phosphorylation of Y^14^ caveolin-1 has been associated with the insulin resistance state in patients with polycystic ovary syndrome [133]. Alternative translation initiation of caveolin-2, a building block of caveolae, may promote the dephosphorylation of INSR by PTP1B and cause insulin resistance [30]. Caveolin-3 knockout mice showed global insulin resistance accompanied by attenuated INSR functions [134].

On the other hand, insulin resistance can distort intracellular activities of INSR, which enters a vicious cycle accelerating the development of metabolic diseases. Chronic hyperinsulinemia resulting from insulin resistance has been shown to decrease insulin-INSR complex internalization and lead to derangements of intracellular receptor trafficking and insulin degradation [135]. Failure on effective insulin clearance dramatically increases insulin action in the short term, whereas in the long term it leads to decreased amounts of INSR on the plasma membrane and contributes to the progression of insulin resistance and further into T2DM. Although INSR endocytosis appears to be significantly associated with insulin resistance, the exact mechanisms remain undefined. Further studies are required to investigate how alternation of INSR endocytosis contributes to the initiation and development of T2DM. 

### 5.3. Disturbance of the INSR Recycling Process: A Possible Initiator of T2DM and a Promising Therapeutic Target

The rates of INSR endocytosis and recycling can regulate the number of available INSR on the plasma membrane [136,137]. Alteration of INSR recycling can lead to insulin resistance and consequently result in hyperinsulinemia and metabolic disorders such as obesity and T2DM. Animal models indicate that insulin resistance in older rats may be caused by a significant decrease in INSR recycling in adipocytes [12]. In human, genetic mutations perturbing endocytosis and recycling have been classified as Class 5 mutations causing insulin resistance [138]. Patients with Glu ^460^ mutation had impaired insulin dissociation from INSR in endosomes, which led to impaired recycling and severe insulin resistance [139]. Defects in INSR internalization and processing were found in monocytes of obese and obese T2DM Patients [140]. A rare INSR genetic mutation on p.Ile119Met, leading to mildly impaired receptor recycling, causes severe insulin resistance in patients [115]. Acidification of the endosomal lumen, due to the presence of proton pumps, results in dissociation of insulin from its receptor [141]. Delayed intracellular dissociation of the insulin-INSR complex was found in cultured Epstein–Barr virus-transformed lymphocytes from patients, whose receptor recycling and insulin processing were impaired [142]. 

Disruption of endocytosis and recycling may alter INSR downstream pathways when INSR is trapped on the endosome. This may contribute to the progression of T2DM. It is reasonable to assume that in an early hyperinsulinemia state such as prediabetes, symptoms are caused by a pseudo-insulin resistance state by impaired INSR trafficking rather than a genuine insensitivity of INSR signaling. The endocytosis and/or recycling process of INSR are slowed down leading to hyperinsulinemia with altered insulin response and insulin clearance. In the meantime, INSR maintains an intact downstream signaling. In a prolonged hyperinsulinemia state, accumulation of unrecycled INSR is targeted for degradation, resulting in deficiency in INSR and impaired insulin clearance ability. Decrease of INSR will eventually lead to cell dysfunction and insulin resistance.

## 6. Conclusions

The importance of INSR trafficking has long been under-appreciated. Evidence has accumulated to suggest the alteration of INSR trafficking in the disease states, especially in metabolic disorders. This ongoing investigation encourages further studies to pursue a new area, namely the interaction between metabolic disorders and INSR trafficking. Although growing evidence underlines the importance of INSR in cell energy metabolism and proliferation in a setting of metabolic diseases, there is still a lack of knowledge on the role of internalization and post-endocytic traffic of INSR in regulating those responses. Comprehending the regulatory mechanisms of INSR trafficking will help to understand the pathogenesis of metabolic diseases, identify potential therapeutic targets, and improve treatment strategies.

## Figures and Tables

**Figure 1 ijms-20-05007-f001:**
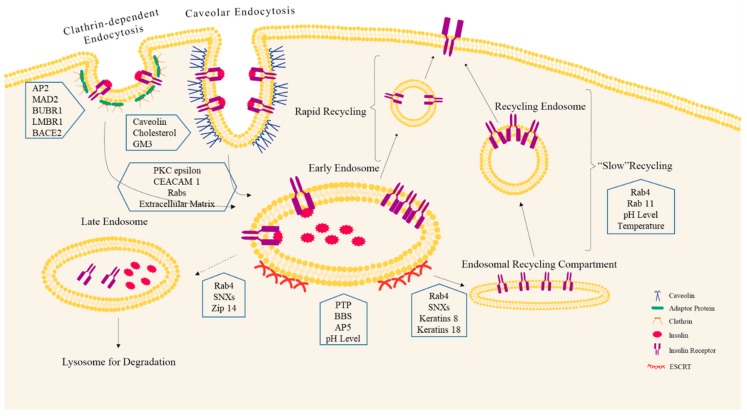
Summary of the existing mechanisms of insulin receptor (INSR) endocytosis, sorting, endosomal trafficking, and recycling. Insulin-INSR complex is internalized via clathrin-dependent and -independent pathways. The complex is broken down in early endosome (EE). Insulin is relocated into late endosome for degradation. INSR is either trafficked to late endosome for degradation or recycled back to the plasma membrane. INSR can be recycled directly from EE via a rapid recycling mechanism or go through the endosomal recycling compartment via a slow recycling process. Proteins involved in different processes are indicated on their corresponding positions.

**Figure 2 ijms-20-05007-f002:**
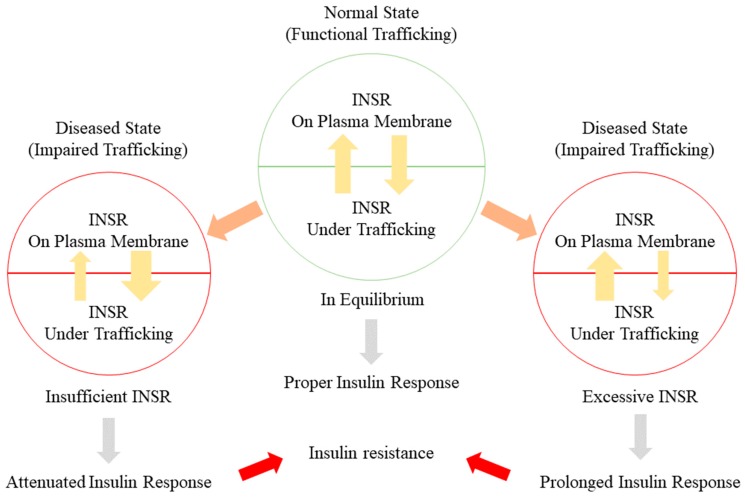
Insulin resistance can be caused by impaired INSR trafficking. The amount of INSR on the plasma membrane and intracellular is maintained in equilibrium. Under normal conditions, a sufficient level of INSR is able to sustain a continuous insulin response under insulin stimulation. In a diseased state when INSR trafficking is damaged, INSR is either trapped intracellularly or restricted on the plasma membrane. Either insufficient or excessive INSR will trigger an altered insulin response and contribute to the development of insulin resistance.

**Table 1 ijms-20-05007-t001:** Mutation of INSR under different conditions with severe insulin resistance.

Condition	Study	Mutation	References
Type A Insulin Resistance Syndrome	Case Study	Missense Mutation	[88,89,90,91,92,93,94,95,96,97]
	Case Study	Splice Site Mutation	[98,99]
	Case Study	Nonsense Mutation	[100,101]
Donohue Syndrome (Leprechaunism)	Case Study	Missense Mutation	[102,103,104,105]
	Case Study	Nonsense Mutation	[106,107,108]
	Regional Study	Frameshift Mutation	[109]
Rabson–Mendenhall Syndrome	Case Study	Missense Mutation	[110,111,112,113,114]
	Regional Study	Missense Mutation	[115]
Familial Hyperinsulinemic Hypoglycemia-5	Case Study	Missense Mutation	[116]
Asymptomatic Hyperinsulinemia	Cohort Study	Nonsense Mutation	[117]
Congenital Muscle Fiber-Type Disproportion Myopathy	Case Study	Missense Mutation	[118]
Non-Insulin Dependent Diabetes Mellitus	Cohort Study	Polymorphism	[119,120,121]

Different mutations of INSR have been collectively summarized [122]. Studies of INSR mutation before 1996 have been summarized and reviewed [123,124].

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
