# Peer review of "Insulin Receptor Trafficking: Consequences for Insulin Sensitivity and Diabetes"

_ijms, 2019, doi:10.3390/ijms20205007_

Round 1

Reviewer 1 Report

This review submitted by Chen et al reports the role of insulin receptor intracellular processes and signaling pathway in normal and diseased states. I think this will give readers a new insight. I would recommend it for acceptance after the minor points.

Comments

I found some of the authors' explanations difficult to follow; I suspect a reader less familiar with the topic might have even greater difficulties. There are many reports about insulin signaling events and diseases in various cell types. In this review, several cell types are listed. You recommend to organizing the topics by cell type or to focusing on some cell types (for example, adipocytes, hepatocytes). And also, it would be easy to understand if there are figures that summarize each item. All the knowledge should be appropriately updated. I think that the authors should look through the literature and update the references and discuss new information based on them.

Author Response

Dear Reviewer,

We really appreciate your time and effort in reviewing this manuscript and providing comments. We have addressed your comments in following dot points.

1. Sentences and language has been thoroughly improved and simplified to explain the concepts clearly and concisely.

2. Proteins and molecules involved in insulin receptor trafficking have been added in figure 1 to address the relevant biological processes. We hope that this addition may help to create a clearer signal pathway to readers.

3. We admit that too many cell types did cause confusion. We have now focused the review on three major cell types: adipocyte and hepatocyte (responding to insulin), and pancreatic beta cell (producing and responding to insulin). The review is restructured around these three cell types. It is hopeful that such restructure shall make the review clearer.  

4. Research on insulin receptor trafficking is progressing slowly after 90s and we have tried our best to gather all new research discoveries since then. The main target of RTK trafficking research was on EGFR for the past decades. We hope to encourage more researchers to shift their focus onto insulin receptor trafficking through this review.

Thank you very much for your constructive comments and suggestions.

Reviewer 2 Report

The authors present a narrative review on the role of insulin receptor trafficking in the pathogenesis of insulin resistance and T2DM.

The review is addressing several major mechanisms in INSR action, from signal binding to endocytosis processes. Most of the paper is based on preclinical in-vitro data, limiting the perspective on clinical outcomes.

Genetic data on humans should be presented more extensively in order to confirm statements from rodent studies and cell models. Parts of chapter 5.2. should therefore be used to form an additional chapter on human genetics, including the already mentioned rare mutations, but also information on case-control studies and GWAS, as far as available. This should be done for all molecules being involved in INSR trafficking and signaling.

Author Response

Dear Reviewer,

We really appreciate your time and effort in reviewing this manuscript and providing comments. We have addressed your comments in following dot points.

1. This review focuses on the trafficking of insulin receptor. Insulin receptor signalling is mentioned as the receptor trafficking process may influence insulin receptor signalling. However, it is not the major target of this review. Detailed insulin receptor signalling is very complex and has been comprehensively reviewed by various researchers.

2. Clinical relevance with diabetes was addressed at the final part of this review. The other parts of the review are dedicated to address the potential molecular regulations behind different stages of the entire insulin receptor trafficking process, which is still distance away from clinical implication yet.

3. More genetic studies are included. We have summarized in new Table 1 about the studies on insulin receptor mutations and the association of these mutations with diseases.

Thank you very much for your constructive comments.

Round 2

Reviewer 2 Report

Paper is ready for publication.